# Molecular Subtypes, Metastatic Pattern and Patient Age in Breast Cancer: An Analysis of Italian Network of Cancer Registries (AIRTUM) Data

**DOI:** 10.3390/jcm10245873

**Published:** 2021-12-14

**Authors:** Giovanna Tagliabue, Sabrina Fabiano, Paolo Contiero, Giulio Barigelletti, Maurizio Castelli, Guido Mazzoleni, Lorenza Boschetti, Anna Clara Fanetti, Antonella Puppo, Antonino Musolino, Claudia Cirilli, Pietro Seghini, Lucia Mangone, Adele Caldarella, Fernanda Lotti, Walter Mazzucco, Andrea Benedetto, Ylenia Maria Dinaro, Ausilia Sferrazza, Pasquala Pinna, Viviana Perotti

**Affiliations:** 1Cancer Registry Unit, Fondazione IRCCS, Istituto Nazionale dei Tumori, 20133 Milan, Italy; sabrina.fabiano@istitutotumori.mi.it (S.F.); giulio.barigelletti@istitutotumori.mi.it (G.B.); viviana.perotti@istitutotumori.mi.it (V.P.); 2Environmental Epidemiology Unit, Fondazione IRCCS, Istituto Nazionale dei Tumori, 20133 Milan, Italy; paolo.contiero@istitutotumori.mi.it; 3Cancer Registry, Aosta Valley Health Authorities Department of Public Health, 11100 Aosta, Italy; mcastelli@ausl.vda.it; 4Cancer Registry, South-Tyrol Local Health Trust, 39100 Bolzano, Italy; GUIDO.MAZZOLENI@sabes.it; 5Cancer Registry, Epidemiology Monitoring Unit, Public Health Agency of Pavia, 27100 Pavia, Italy; Lorenza_Boschetti@ats-pavia.it; 6Sondrio Cancer Registry, Health Protection Agency, 23100 Sondrio, Italy; ac.fanetti@ats-montagna.it; 7Clinical Epidemiology Unit, Liguria Cancer Registry, IRCCS-Ospedale Policlinico San Martino, 16132 Genova, Italy; antonella.puppo@hsanmartino.it; 8Department of Medicine and Surgery, University of Parma, Medical Oncology, Cancer Registry, University Hospital of Parma, 43100 Parma, Italy; amusolino@ao.pr.it; 9Local Health Unit, 41100 Modena, Italy; c.cirilli@ausl.mo.it; 10Cancer Registry, Department of Epidemiology, Piacenza General Hospital, 29121 Piacenza, Italy; p.seghini@ausl.pc.it; 11Epidemiology Unit, AUSL-IRCCS di Reggio Emilia, 42121 Reggio Emilia, Italy; lucia.mangone@ausl.re.it; 12Institute for Cancer Research, Prevention and Clinical Network (ISPRO), 50139 Florence, Italy; a.caldarella@ispro.toscana.it; 13Section of the Puglia Cancer Registry, Cancer Registry, Local Health Unit Brindisi, 72100 Brindisi, Italy; fernanda.lotti@asl.brindisi.it; 14Department of Sciences for Health Promotion and Mother and Child Care “Giuseppe D’Alessandro”, University of Palermo, 90128 Palermo, Italy; walter.mazzucco@unipa.it; 15Integrated Cancer Registry of Catania-Messina-Enna, Department of Hygiene and Public Health, 95100 Catania, Italy; studiogeo.benedetto@gmail.com; 16Siracusa Cancer Registry, Health Unit of Siracusa, 96100 Siracusa, Italy; ylenia.dinaro@asp.sr.it; 17Ragusa Cancer Registry, Provincial Health Unit, 97100 Ragusa, Italy; ausilia.sferrazza@asp.rg.it; 18Nuoro Cancer Registry, ASSL Nuoro/ATS Sardegna, 08100 Nuoro, Italy; pasquala.pinna@atssardegna.it

**Keywords:** hormone receptor (HR), human epidermal growth factor receptor 2 (HER2), breast cancer, metastasis, age, population-based, cancer registry

## Abstract

Breast cancer stage at diagnosis, patient age and molecular tumor subtype influence disease progression. The aim of this study was to analyze the relationships between these factors and survival in breast cancer patients among the Italian population using data from the AIRTUM national database. We enrolled women with primary breast cancer from 17 population-based cancer registries. Patients were subdivided into older (>69 years), middle (50–69 years) and younger age groups (<50 years) and their primary tumors categorized into four molecular subtypes based on hormone receptor (HR) and human epidermal growth factor receptor 2 (HER2) status. There were 8831 patients diagnosed between 2010 and 2012 included. The most represented age group was 50–69 years (41.7%). In 5735 cases the molecular subtype was identified: HER2–/HR+ was the most frequent (66.2%) and HER2+/HR− the least (6.2%). Of the 390 women with metastases at diagnosis, 38% had simultaneous involvement of multiple sites, independent of age and molecular profile. In women with a single metastatic site, bone (20% of cases), liver (11%), lung (7%) and brain (3%) were the most frequent. In the studied age groups with different receptor expression profiles, the tumor metastasized to target organs with differing frequencies, affecting survival. Five-year survival was lowest in women with triple-negative (HER2−/HR–) tumors and women with brain metastases at diagnosis.

## 1. Introduction

Breast cancer is the most frequent cancer type among women: the estimated incidence in Italy in 2020 was just under 55,000 cases, with an estimated five-year survival rate of more than 87% [1]. Breast cancer is also the neoplastic disease with the highest prevalence among women: 43% of Italian women live with a breast cancer diagnosis [1]. Furthermore, breast cancer is the first cause of death among women. In Italy, differences in incidence have been observed between the north (highest, 162.6 cases per 100,000 women), center (145.2 cases per 100,000 women), and south and islands (lowest, 123.6 cases per 100,000 women) [1]. Although the prognosis is favorable in most cases, about 5–8% of women have distant metastases at the time of diagnosis and the five-year cause-specific survival for these patients is very low, ranging from 24% to 39% [2]. In fact, 90% of breast cancer deaths are attributable to metastases occurring during treatment [3].

Tumor stage at diagnosis, patient age and molecular subtype may influence disease progression. Four molecular subtypes have been identified in tumor tissue based on the degree of expression of human epidermal growth factor 2 (HER2) and hormone receptors (HR, including estrogen receptors [ER] and progesterone receptors [PR]): HER2−/HR+, HER2+/HR+, HER2+/HR− and HER2−/HR− (also referred to as triple-negative) [4]. The different molecular profiles expressed by tumor cells are used for prognostic and treatment-guidance purposes: HR+ cells respond better to hormonal therapies, while in HER2+ breast cancer neoadjuvant therapy has become a commonly used option. Triple-negative tumors (HER2−/HR−) have the worst prognosis, showing aggressive histological features, unresponsiveness to the common endocrine therapies and shorter survival [5]. Furthermore, HER2+ tumors are considered highly aggressive, with high mortality [6,7]; however, the HR+/HER2+ subtype has recently gained a better prognosis particularly in metastatic tumors, because it has molecular targets for hormone therapy as well as other targeted treatments such as trastuzumab [8,9,10]. Patients with different molecular subtypes have shown different response rates even to established therapies such as chemotherapy and radiotherapy, influencing disease recurrence and survival [11].

Age at diagnosis appears to play an important role in breast cancer prognosis [12], but few studies have focused on the role of age in women with metastatic disease. Previous investigations revealed that in younger patients breast cancer may present a more aggressive biological behavior [13], while in elderly patients triple-negative breast cancer is associated with higher mortality than in the younger cohort in the first two years after diagnosis [14].

In Italy, a national network of cancer registries, AIRTUM, has been active since the late 1990s. It covers almost the entire country [15] and systematically records all cases of cancer arising in the Italian population. For the present study, we decided to analyze the molecular subtype of tumors as reported by pathologists. Unlike hospital-based observational systems, where clinical detail is the strong point, a population-based registry provides a general view of the disease in the source population, which is extremely heterogeneous and should be free of any selection bias.

The objective of the study was to analyze the relationships between patient age, molecular subtype, sites of metastasis and survival in women with breast cancer in the Italian population, using data from the AIRTUM national database. Our ultimate aim was to gain a better understanding of breast cancer development and to establish clinical pathways for the management of metastatic diseases that allow us to improve the lives of these patients.

## 2. Materials and Methods

### 2.1. Study Design

This is a population-based study analyzing data from a network of 50 general cancer registries (for all cancers) and seven specialized registries (for specific age groups or cancer sites).

### 2.2. Data Sources

AIRTUM collects data from these registries and, after validating their quality and completeness according to the quality checks of the International Agency for Research on Cancer (IARC) and the International Association of Cancer Registries (IACR), uses them for collaborative studies in cancer epidemiology research.

All incident cases of cancer registered among Italian residents on the basis of regional mortality, pathology, laboratory, clinical and hospital discharge reports are sent by the registries to the AIRTUM database. Cases are coded according to the International Classification of Diseases for Oncology, third edition (ICD-O-3) [16] and staged according to the TNM classification, 6th and 7th editions [17,18]. For the present study, cases were categorized into four stages (I, II, III and IV).

A case series was compiled by selecting primary breast cancers according to ICD-O-3 classes C50.0–C50.9 with an invasive morphology. Patients were excluded from the analysis if the diagnosis was derived from a death certificate or autopsy report. For our analysis we grouped the case series into three age ranges at diagnosis: women younger than 50 years, women between 50 and 69 years, and women older than 69 years.

Seventeen registries participated in the study: 14 provided incident cases for 2011, two (Tuscany and Brindisi) provided data covering 2010, and the remaining registry (Reggio Emilia) provided data for 2012.

The following variables were collected for each patient: age at diagnosis, tumor site, tumor morphology, stage at diagnosis, metastatic site, hormone receptor status (ER, PR), HER2 expression, date and status at follow-up and, in the case of patient death, date and cause of death.

We categorized breast cancer cases into four molecular subtypes based on hormone receptor status and HER2 expression. ER, PR and HER2 status were recorded by the registries according to the pathologist’s interpretation of the assays. In the analysis of receptor expression, we excluded patients whose receptor status was unknown (either ER and PR or HER2 unknown) or incomplete (only one receptor recorded).

### 2.3. Statistical Analysis

Descriptive statistics were used to examine the baseline characteristics of breast cancer patients. The significance of differences in distribution frequencies between the analyzed categories was assessed by means of the chi-square test with R Studio, version 3.2.5 [19]. A two-sided *p*-value less than 0.05 was considered statistically significant. Incidence rates (per 100,000 per year, unless otherwise stated) were age-standardized using direct methods and the world standard population 2000–2025 [20]. Rates were estimated using the SEER*Stat statistical software from Surveillance Epidemiology and End Results (SEER) [21]. Five-year relative survival estimates were obtained considering the pool of 17 registries with cases diagnosed between 2010 and 2012, and followed up to 31 December 2017. Relative survival is defined as the ratio of the observed survival to the expected survival in the general population of the same age and sex; it is used to correct for deaths from causes other than the cancer under investigation. Relative survival was calculated for patients aged 0–99 years by means of the Stata software package, version 16 (StataCorp LLC, Release 16. College Station, TX, USA) [22].

### 2.4. Outcome and Follow-Up

Passive and active monitoring of cancer cases was carried out from the date of diagnosis to the end of follow-up (31 December 2017), when the patients’ vital status was ascertained. The outcome variables were: alive at end of follow-up; deceased including date of death of any cause; and censored due to loss or incomplete follow-up. The data were obtained through record linkage with Local Health Authority registries (listing all persons eligible for health care) and mortality registries.

## 3. Results

### 3.1. Setting

Seventeen population-based cancer registries participated in the study, for a total observed population of 5,571,994. In the study period, 8831 cases of primary invasive breast cancer were identified; in 93% there was microscopic confirmation of the tumor. The participating registries were spread over Italy. The south and islands had the largest number of cases (3089 patients; 35%). The highest incidence rates per 100,000 were registered in Modena (98.5), Pavia (96.3) and Parma (95.5); the lowest in Ragusa (63.5), Palermo (71.4) and Nuoro (73.4) (Appendix A).

Table 1 summarizes the frequency and proportions of the studied characteristics among the patients. There were significant differences in tumor stage, metastases, tumor type and receptor status (*p* < 0.001) between the different age groups. The median age of the women was 61 years. Overall, 402 patients (4.6%) had metastases at diagnosis. At the end of 2017, 1856 patients had died (all-cause mortality 21%): 171 (9%) in the <50-year age group, 465 (26%) in the 50–69-year age group and 1220 (66%) in the >69-year age group. Most of the women in all age groups were diagnosed with early-stage cancer (stage I and II), with the highest incidence observed in women aged 50–69 years. The percentage of metastatic disease at diagnosis was highest in the oldest age group, as was the percentage of cases with unknown disease stage at diagnosis.

The most common histological type was ductal carcinoma (72.9%), followed by lobular carcinoma (14.9%) and other morphologies (6.2%). In 6% of cases tumor morphology was classified as non-specific (ICD-O-3 code 8000/3, malignant neoplasm or 8010/3, carcinoma), as can be expected in a population-based sample.

### 3.2. Molecular Subtypes

Of the analyzed receptors, ER was most frequently expressed in our patient series, followed by PR and HER2 (Figure 1).

The women with a complete receptor profile were 5735 of a total of 8831 (64.9%). Molecular subtypes were reconstructed using the available information on hormone receptors and HER2 expression. The most frequent subtype among these 5735 patients was HER2−/HR+, whereas HER2+/HR− was the least frequent (Figure 2).

When receptor expression was analyzed by age, the most frequent molecular subtype in all three groups was HER2−/HR+, ranging from 61.8% to 69.8% with increasing age. The HER2+/HR− subtype decreased in frequency from 7.8% to 4.7% with increasing age and the HER2+/HR+ type from 20.6% to 17.1%. The triple-negative subtype (HER2−/HR−) occurred in a range of 8–10% (Table 1).

Reconstructing the molecular subtype requires information on the expression of all three receptors (ER, PR and HER2); the proportion of cases not typed because part or all of the information was missing ranged from 7.2% to 13.8% in the three age groups (Table 1).

### 3.3. Sites of Metastasis at Diagnosis

The analysis of metastatic sites at diagnosis was performed considering patients of the three age groups. In the total case series, women with metastatic disease at diagnosis were 402 (4.6% of 8831 women); in the group of patients with molecular typing, cases with metastases were 390 (6.8% of 5735).

In this analysis, as shown in Figure 3, metastases arising at the primary target sites of breast cancer metastasis (bone, liver, lung and brain) were taken into account; combinations of these metastatic sites, as well as less frequently involved sites (specifically pleura, skin, adrenal glands, non-locoregional lymph nodes and digestive tract), were also considered.

In Table 2 the metastatic sites are broken down by age group. Excluding patients with multiple metastases (more than one involved site), we see that bone was a frequent site in all three age groups, while the brain was affected mostly in younger women. Liver or lung metastases at diagnosis were found mainly in the oldest age group.

Patients with simultaneous metastases at multiple sites generally have a poorer prognosis [23]. In our study, multiple metastatic sites were present in 38% of the overall group. The highest percentage (44.3%) was in women younger than 50 years; it decreased progressively with increasing age (Table 2).

It should be noted that in 5% of women with a complete molecular profile and metastatic disease at diagnosis, the site of metastasis was not reported by the registries. 

The relationship between molecular subtypes, patient age and sites of metastasis was analyzed only for the most frequently involved sites (Figure 4).

In patients younger than 50 years, 19.4% of those with a HER2−/HR+ molecular subtype had multiple-organ metastases, followed by metastases to bone (9.7%) or liver (5.6%). Lung metastases were second in frequency (2.8%) in the HER2+/HR+ subtype. In patients with triple-negative tumors (HER2-/HR-), multiple-organ metastases at diagnosis were the most frequent (11.1%), followed by bone metastases (5.6%). HER2+/HR− patients presented metastases at multiple sites in 6.9% of cases; metastases to the liver or brain were present in 1.4%.

In women aged between 50 and 69 years with a HER2−/HR+ molecular subtype, metastases to multiple organs (16.8%) and bone (14.5%) were the most frequent, followed by metastases to the liver (5.2%) or brain (1.7%); women with a triple-negative subtype had metastases to bone or liver in 2.9% and 2.3% of cases.

The four molecular subtypes in this age group were associated with metastatic disease at more than one site in 16.8%, 11.6%, 6.9% and 6.4%, respectively.

In patients older than 69 years with a HER2−/HR+ molecular profile, liver (6.9%), lung (6.9%), bone (15.2%) and multiple organs (16.6%) were the most frequent metastatic sites at diagnosis. Patients with a triple-negative molecular subtype presented metastases to multiple sites or to the lung in 3.4% of cases; metastases to the liver (2.8%) and bone were present in 2.8% and 1.4%, respectively. 

In both younger and older patients, the most frequently metastatic molecular subtype at diagnosis was HER2−/HR+, which was also the most represented subtype in our case series.

### 3.4. Survival

Of the 8831 patients in the study, 1856 (21%) died during the study period, 967 of breast cancer and 889 of other causes. The five-year survival curves for women with metastatic cancer and molecular typing show that a triple-negative molecular profile (HER2−/HR−) was associated with the worst survival, while the best survival was seen in women with HER2−/HR+ or HER2+/HR+ tumors (Figure 5). In Appendix A we showed the relative survival by stage.

Analyzing survival according to the main site of metastasis (Figure 6), we observed that, regardless of the molecular subtype, women with brain metastases at diagnosis had the worst survival. Overlapping survival rates were seen in women with metastases to the lung or liver at diagnosis, while the survival rate was highest in patients with bone as the main site of metastasis. The survival advantage was still present, albeit small, five years after diagnosis.

## 4. Discussion

### 4.1. Incidence Rate

The results of the study confirmed the different incidence rates of invasive breast cancer in the three geographical areas considered, with higher rates in central and northern Italy than southern Italy, a finding consistent with data from the literature [24] (Appendix A).

### 4.2. Age at Diagnosis

Almost half of breast cancers in our study occurred in women aged between 50 and 69 years (41.7%); older women accounted for 36% of the case series and younger women for 22.3%. The distribution of cases by age was very similar to that described by Auguste [25], except for the oldest age group (over 74 years, 21.4% of cases). Age is to be considered a risk factor for breast cancer. In a study by Roder et al., of 493 patients in Australia with breast cancer diagnosed between 1998 and 2005, women under 40 years and those over 70 years of age had worse overall survival rates than women aged 40–69 years [26]. A Swedish study of 4453 breast cancer patients diagnosed between 1961 and 1991 at a single institution and followed up for 10 years found that women under 40 and those over 80 years had higher breast–cancer-specific mortality and a poorer prognosis than those in other age groups [27]. Younger patients have also been found more likely to be HR-negative and elderly patients more likely to be HR-positive [28]. In our study, 17.6% of young women were HR-negative versus 14.4% in the middle age group and 13% in the older age group; it should be noted, however, that receptor expression data were missing in a proportion of cases (7.2%, 14.1% and 13.8% in the three age groups).

### 4.3. Stage at Diagnosis

Most women in our case series had non-metastatic cancer when they were diagnosed (stage I, II or III). This is partly attributable to the fact that the most represented age group in our study, women aged 50–69 years (3679 cases), undergo routine population screening, which detects breast cancer at its earliest stages. Only 4.6% of our overall series presented metastatic disease at diagnosis.

Similar numbers were reported by others. In the study by Gong et al. [2], based on data from SEER registries, the percentage of women with metastatic cancer at diagnosis was 4.8%, while Wang et al. [29] reported metastases at diagnosis in 4.9%.

In our study, 19.9% of cases lacked information on the stage at diagnosis. Allemani et al., reporting on breast cancer survival in the US and Europe in the CONCORD cohort, listed missing stages in only 8% of European registries overall, but up to 18–22% in some countries, notably Finland and Italy [30].

The presence of a significant proportion of cases without information on the disease stage is a limitation of our study, but since this is a population-based study, we used the data made available by the registries.

### 4.4. Molecular Subtypes

A number of molecular subtypes of breast cancer have been identified that have prognostic significance and can be used as targets in personalized medicine [31]. Hormone receptors and HER2 are the biomarkers of choice for decision-making in breast cancer, as their expression affects both prognosis and treatment. Vuong and colleagues’ overview of the molecular classification of breast cancer suggests that 80% of cases can be expected to be positive for ER and 13–20% for HER2, while approximately 10–15% have a triple-negative molecular profile [32].

The percentage of HER2−/HR+ cases in our study increased from 61.8% in younger women to 65.8% and 69.8% in women in the middle and older age groups. These data are comparable to those reported by Auguste et al. [25], Press et al. [33] and Abdel-Rahman [34]. Press and coworkers reported higher percentages of women with a HER2−/HR+ molecular subtype: 64.2%, 72% and 77.7% in the age groups below 50, between 50 and 70, and above 70 years, respectively, but their case series was fully typed, while molecular typing was unavailable in approximately 12% of our series.

A triple-negative profile was identified in 11.7%, 7.6% and 8.8% of the women in the three age groups of the Auguste study (<50, 50–74 and >74 years). In our study cohort, these percentages were 9.8%, 8% and 8.3%, respectively. There was a similar decrease in the percentage in the middle group, although the difference in grouping in the two studies must be taken into account.

The Italian study by Caldarella et al., [7] on a cohort of women belonging to the same geographic area as those in our study did not group patients by age and found HER2−/HR+ expression in 70.3% of patients, HER2+/HR+ in 15.6%, HER2+/HR− in 6%, and HER2−/HR− in 8.1%, results superimposable to ours if we compare the patients as a whole (66.2%, 19.1%, 6.2% and 8.5%).

### 4.5. Metastases

This study focuses on the relationships between pattern of metastasis, molecular subtypes of breast cancer and age at diagnosis. The results indicate that in the age groups considered the tumor metastasized to target organs with a differing frequency depending on the receptor expression profile, which had an impact on survival. Patients with bone metastases showed the best prognosis and patients with brain metastases had an unfavorable prognosis, also because treatment options for brain metastasis are severely limited [34].

Bone is one of the sites most susceptible to tumor spread, especially in breast cancer [7]. In our study, 25.1% of patients overall (22.8% in the younger group, 19.9% in the middle-age group and 30.4% in the older group) had bone metastases. In the study by Xiong et al. [35], 62.5% of patients with metastatic breast cancer at diagnosis had bone metastases, while Soni et al. [36] reported bone metastases in 48% of patients. The lower percentage in our study is probably due to the lack of information about the site of metastatic onset in a considerable proportion of cases with metastases at diagnosis.

HR+ tumors seem to have a propensity to spread to the bone. The mechanism behind this is unknown, although some studies suggest that tumor dormancy may play a role [37].

Another frequent site of breast cancer metastases is the lung. Previous studies reported that the likelihood of tumor spread to the lung is higher in patients with a HER2+/HR+ or HER2−/HR− molecular subtype [38]. In the study by Xiao et al. [38] 10.5% of patients had metastases to the lung, while in our study lung metastases were present in 7% of patients, in particular those with HER2−/HR+ tumors, regardless of age. In the paper by Gong et al. [2] the percentage of patients with metastases to the lung was 16%; broken down by molecular subtype, the percentage was lower (11%) in women with HER2−/HR+ tumors and conspicuously higher (33.1%) in those with HER2−/HR− tumors.

HER2 status has been reported to have a strong relationship to brain metastasis. Patients with HER2−/HR+ and HER2+ tumors are at high risk of brain metastases [39], and those with a HER2−/HR- subtype have a significantly higher probability of brain metastasis than those with other molecular subtypes [40]. Studies found that patients with HER2−/HR− tumors present relatively higher expression of EGFR, and EGFR increases the risk of brain metastases in breast cancer patients [41]. In our study, brain metastases were present in 2.6% of patients, mostly with a HER2+/HR− molecular subtype in the two younger age groups.

### 4.6. Survival

Of the 1856 deceased study patients, 52% died of breast cancer and 48% of other causes. Relative survival was estimated at five years for all registries but three, which provided four-year follow-up data.

Previous studies have shown that survival differences in female breast cancer may be linked to different metastatic sites [2,42], survival being higher in patients with bone metastasis and lower in the case of brain metastasis; this corresponds to our own findings. Similarly to a study published by SEER [23], in our study patients with only lung or only liver involvement had similar median survival times.

We found that patients with HER2−/HR+ tumors had the best survival, whereas survival was poorest in those with a HER2−/HR− molecular profile, as also reported by others [33].

### 4.7. Strengths and Limitations

A limitation of our study is that registry data on tumor stage or receptor status were incomplete in a considerable proportion of cases and therefore could not be used. Despite this limitation, the data resulted in an interesting analysis, allowing us to correlate molecular profiles with disease characteristics in patients at different stages of life and in different regions and clinical situations.

Another aspect that would have made the story of the disease under study more complete concerns the treatment paths undertaken by the patients, but the experience of large population-based studies (EUROCARE) has taught us that collecting these variables on a population level requires ad hoc studies [43].

A strength of our analysis is that it was based on data routinely collected by registry operators, showing the feasibility of carrying out molecular typing studies without the effort and cost of high-resolution studies.

## 5. Conclusions

This epidemiological study describes breast cancer in the Italian population using variables collected from population registries. It characterizes breast tumors on the basis of their molecular profile (defined by the expression of hormone receptors and HER2) and the occurrence of metastases at diagnosis in relation to patient age. Our analysis revealed interesting differences pertaining to these characteristics, for example in bone metastases for patients with a HER2−/HR+ molecular profile in the younger age group with respect to the oldest group. We hope the information presented in this paper will help to gain a deeper understanding of breast cancer development and to establish clinical pathways for the management of metastatic diseases that allow us to improve the lives of these patients.

## Figures and Tables

**Figure 1 jcm-10-05873-f001:**
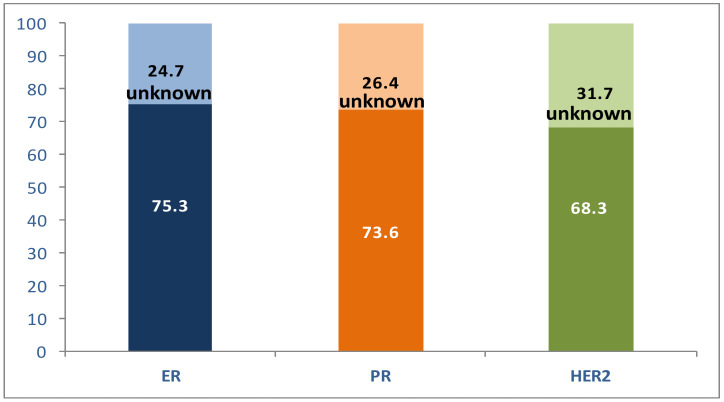
Distribution of receptor status in the study patients (%).

**Figure 2 jcm-10-05873-f002:**
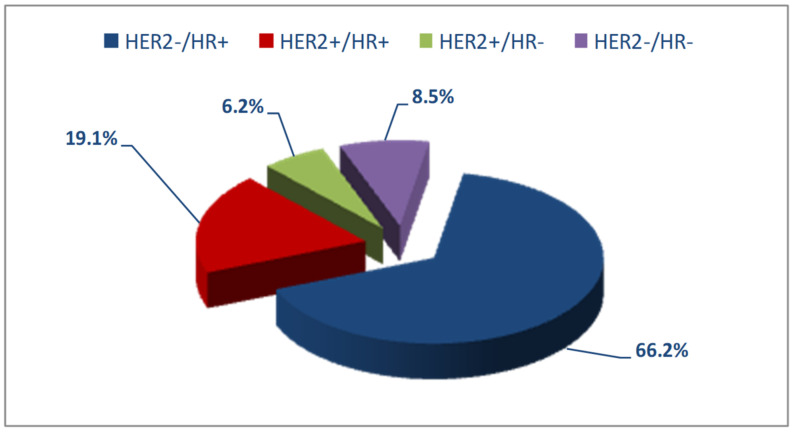
Distribution of molecular subtypes in the study patients.

**Figure 3 jcm-10-05873-f003:**
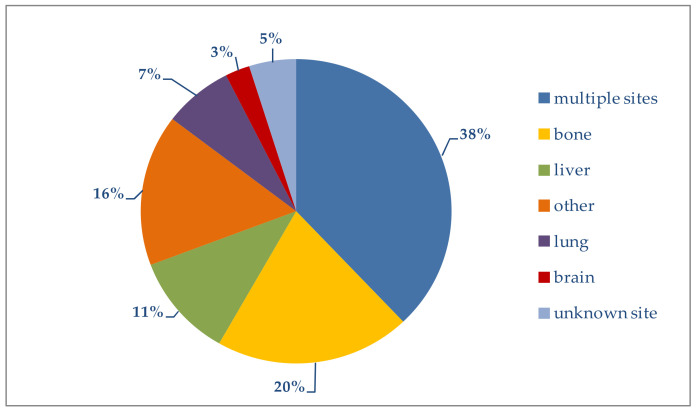
Distribution of metastatic sites.

**Figure 4 jcm-10-05873-f004:**
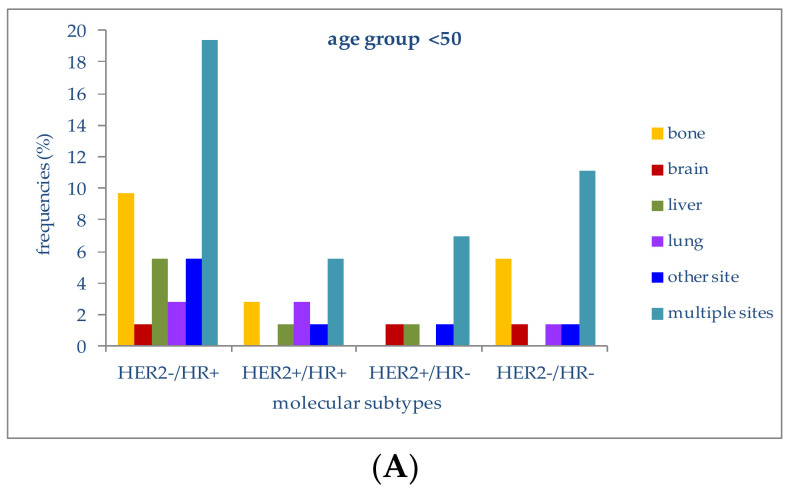
(**A**) Frequencies of metastatic sites in stage IV patients with different molecular subtypes in the age group <50 years; (**B**) Frequencies of metastatic sites in stage IV patients with different molecular subtypes in the age group 50–69 years; (**C**) Frequencies of metastatic sites in stage IV patients with different molecular subtypes in the age group >69 years.

**Figure 5 jcm-10-05873-f005:**
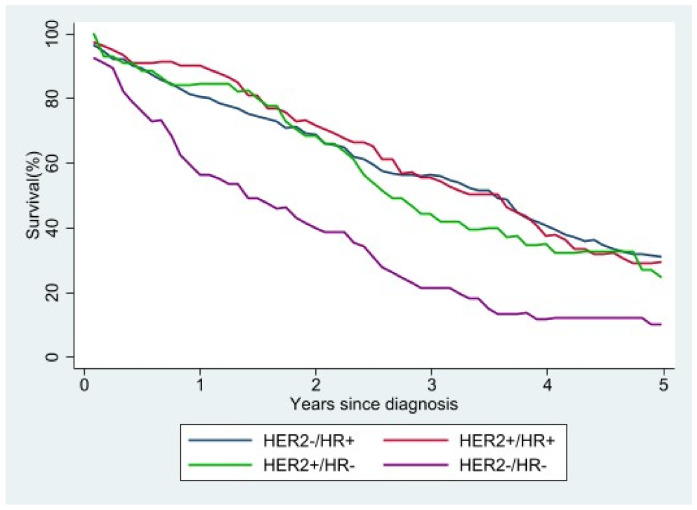
Relative survival of stage IV patients according to molecular subtype.

**Figure 6 jcm-10-05873-f006:**
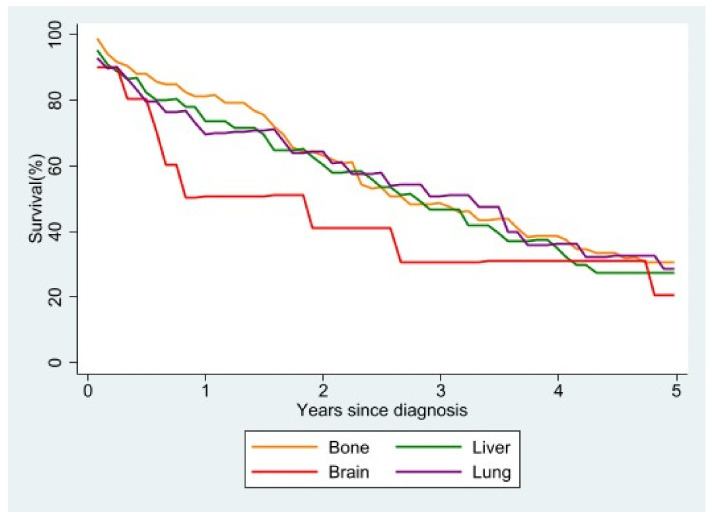
Relative survival of stage IV patients with single metastasis at diagnosis according to site of metastasis.

**Table 1 jcm-10-05873-t001:** Characteristics of breast cancer patients according to age group.

Age (Years)		<50	50–69	>69	
Total Patients		n = 8831			
		n (%)	n (%)	n (%)	*p*-Value *
Number		1969 (22.3)	3679 (41.7)	3183 (36.0)	<0.001
Stage (TNM)	I	730 (37.1)	1633 (44.4)	840 (26.4)	<0.001
	II	612 (31.1)	932 (25.3)	793 (24.9)	
	III	278 (14.1)	445 (12.1)	407 (12.8)	
	IV	57 (2.9)	161 (4.4)	184 (5.8)	
	Unknown	292 (14.8)	508 (13.8)	959 (30.1)	
Metastasis	No	1375 (69.8)	2456 (66.8)	1909 (60.0)	<0.001
	Yes	57 (2.9)	161 (4.4)	184 (5.8)	
	Unknown	537 (27.3)	1062 (28.9)	1090 (34.2)	
Tumor histology	Ductal	1540 (78.2)	2773 (75.4)	2125 (66.8)	<0.001
	Lobular	248 (12.6)	596 (16.2)	473 (14.9)	
	Other	92 (4.7)	143 (3.9)	315 (9.9)	
	NOS	89 (4.5)	167 (4.5)	270 (8.5)	
Receptor expression	HER2-/HR+	826 (61.8)	1603 (65.8)	1369 (69.8)	<0.001
	HER2+/HR+	275 (20.6)	482 (19.8)	336 (17.1)	
	HER2+/HR−	104 (7.8)	157 (6.4)	93 (4.7)	
	HER2−/HR−	131 (9.8)	196 (8.0)	163 (8.3)	
	Unknown	633 (7.2)	1241 (14.1)	1222 (13.8)	

* *p*-values (chi-square test) for differences between subgroups. TNM—tumor-node-metastasis; NOS—not otherwise specified; HER—human epidermal growth factor receptor; HR—hormone receptor.

**Table 2 jcm-10-05873-t002:** Number and proportion of breast cancer patients with single and multiple sites of metastasis.

Age (Years)	<50		50–69		>69				
	n	%	n	%	n	%	Total	%	*p*-Value *
**Multiple sites**	31	44.3	72	41.9	45	30.4	148	**37.9**	<0.001
**Bone**	13	18.6	34	19.8	32	21.6	79	**20.3**	0.006
**Liver**	6	8.6	19	11.0	18	12.2	43	**11.0**	0.025
**Other**	7	10.0	32	18.6	24	16.2	63	**16.2**	<0.001
**Lung**	3	4.3	5	2.9	20	13.5	28	**7.1**	<0.001
**Brain**	3	4.2	4	2.3	3	2.0	10	**2.6**	0.904
**Unknown site**	7	10.0	6	3.5	6	4.1	19	**4.9**	0.948

* *p*-values (chi-square test) for differences between subgroups.

## Data Availability

The data on breast cancer cases used in this study were provided by cancer registries affiliated to AIRTUM and cannot be made freely available.

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
