# Peer review of "Molecular Subtypes, Metastatic Pattern and Patient Age in Breast Cancer: An Analysis of Italian Network of Cancer Registries (AIRTUM) Data"

_jcm, 2021, doi:10.3390/jcm10245873_

Round 1
Reviewer 1 Report
A very nice, well conducted study of breast cancer patients recorded in Italian Registries. I have the following comments and recommendations.
1.) Materials and Methods. At what point during the course of the disease was the data entered. Was data entered at more than one time point for patients.
2.) What were the criteria for HER2 positivity?
3.) Results, 3.1, line 3. How was data from the other 7% without histologic confirmation managed.
4.) Results, section 3.1. A difference in the distribution of cases according to geographic region was noted. Was this in fact a geographical difference, that is, a difference due to environmental or other factors, or was this due to a difference in documentation and registration of cancer cases between regions. It is mentioned later that data for tumor stage and receptor was incomplete between registries. Did the same apply to collection of cancer cases?
5.) The nature of the comparisons in Table 1 are unclear. Are the comparisons for each category between age groups, between elements of each category within age groups, or both?
6.) Is the designation of metastases for distant metastases only, or are lymph node metastases included?
7.) It would be interesting, and quite helpful, to present survival according to stage to provide another important parameter in addition to subtype, especially as this is a very well documented patient population. This allows comparison with many studies in the past to indicate improved prognosis over time for each stage of breast cancer.
8.) What is the median follow-up for the survival studies.
9.) Conclusions. This is a very well done and important study of breast cancer in Italy. As the authors note, there are other studies from registries in other countries for comparison. It would be helpful to indicate how this study adds to other studies for our understanding of the presentation and course of breast cancer, and how this might be used to establish clinical pathways for the management of metastatic disease (as well as other stages of breast cancer).
Author Response
We would like to thank the reviewer for reviewing our work and for their constructive suggestions.
We have modified the text in accordance with the suggestions by reviewer.
https://susy.mdpi.com/user/manuscripts/review/22967426?report=16300475
Reviewer 1
(x) I would not like to sign my review report
( ) I would like to sign my review report English language and style
( ) Extensive editing of English language and style required
( ) Moderate English changes required
(x) English language and style are fine/minor spell check required
( ) I don't feel qualified to judge about the English language and style
|
Yes |
Can be improved |
Must be improved |
Not applicable |
|
|
Does the introduction provide sufficient background and include all relevant references? |
(x) |
( ) |
( ) |
( ) |
|
Is the research design appropriate? |
(x) |
( ) |
( ) |
( ) |
|
Are the methods adequately described? |
(x) |
( ) |
( ) |
( ) |
|
Are the results clearly presented? |
(x) |
( ) |
( ) |
( ) |
|
Are the conclusions supported by the results? |
(x) |
( ) |
( ) |
( ) |
Comments and Suggestions for Authors
A very nice, well conducted study of breast cancer patients recorded in Italian Registries. I have the following comments and recommendations.
1.) Materials and Methods. At what point during the course of the disease was the data entered.
Population cancer registries use information source data such as medical records, pathology reports and hospital discharges collected at the time of diagnosis and treatment.
2.) What were the criteria for HER2 positivity?
Cancer registries use pathological reports as an important source of information. Each registry has data from the pathological services present on the area covered by the registration.
Pathologists in the measure of diagnostic tumor biomarkers use internationally standardized procedures which makes the data comparable. Pathologists of the registries' reference services send to the cancer registries the criteria they use to establish marker positivity/negativity thresholds. Based on these guide line are operated conversions that allow to make homogeneous the values found.
3.) Results, 3.1, line 3. How was data from the other 7% without histologic confirmation managed.
In studies using population-based data, it is accepted that case studies present information for clinical variables there are not mandatory for cancer registration with a lower degree of completeness than that contained in a clinical dataset. These cases without morphological information were taken into account in the analysis. Cases without morphological information, because the register failed to access the complete morphological information, are recorded as cases without morphology , or NOS cases (not otherwise specified).
4.) Results, section 3.1. A difference in the distribution of cases according to geographic region was noted. Was this in fact a geographical difference, that is, a difference due to environmental or other factors, or was this due to a difference in documentation and registration of cancer cases between regions. It is mentioned later that data for tumor stage and receptor was incomplete between registries. Did the same apply to collection of cancer cases?
There is a known gradient of cancer incidence between macro-areas of northern Italy and southern associated with different factors such as lifestyles (smoking, diet) and pollution or other factors: for example, a higher incidence of liver cancer was related to local genetic and environmental conditions such as prevalence of hepatitis B/C virus infection, peculiar in the regions of Southern Italy (https://www.registri-tumori.it/cms/pubblicazioni/i-numeri-del-cancro-italia-2019).
Population cancer registry data follow rules for cancer registration and coding and for statistical analysis for cancer registries (Cancer Registration: Principles and Methods. IARC Scientific Publication No. 95, Edited by Jensen OM, Parkin DM, MacLennan R, Muir CS, Skeet RG).
Methodologically, a high standardization of the data collected has been achieved in Italy thanks to the presence of a national network (AIRTUM). AIRTUM coordinates and supports the activity of cancer registration, standardises recording techniques, encourages joint analysis of data and contributes, on the basis of scientific expertise, the planning of new registration initiatives and their evaluation.
5.) The nature of the comparisons in Table 1 are unclear. Are the comparisons for each category between age groups, between elements of each category within age groups, or both?
Table 1 summarizes the frequency and proportions of the studied characteristics among the patients. There were significant differences in tumor stage, metastases, tumor type and receptor status (p<0.001) between the different age groups.
6.) Is the designation of metastases for distant metastases only, or are lymph node metastases included?
Metastasis at diagnosis are distant metastasis, excluded loco-regional lymph nodes.
7.) It would be interesting, and quite helpful, to present survival according to stage to provide another important parameter in addition to subtype, especially as this is a very well documented patient population. This allows comparison with many studies in the past to indicate improved prognosis over time for each stage of breast cancer.
We added, according to referee suggestion, a new patient survival graph by stage at diagnosis (supplementary figure S2).
8.) What is the median follow-up for the survival studies.
Based on our data, we have calculated that the median survival time in patients with metastases at diagnosis is 2.5 years.
9.) Conclusions.
This is a very well done and important study of breast cancer in Italy. As the authors note, there are other studies from registries in other countries for comparison. It would be helpful to indicate how this study adds to other studies for our understanding of the presentation and course of breast cancer, and how this might be used to establish clinical pathways for the management of metastatic disease (as well as other stages of breast cancer).
We thank the reviewer for the important comment that we led us to clarify the aim of the study.
Compared to other published studies on the breast, which examine the relationships between molecular profile, metastatization, survival, age at diagnosis, our study analyzed in a joint way both the age of patients at diagnosis , both receptor profile and location of metastases in a population context: the results indicate how the receptor profile, combined with the age of onset of the disease may be a valid prognostic indicator and predictor of onset of metastasis in certain target organs. In addition, the population context provides a large number of cases on a wide territory, such as Italy.
Submission Date
22 October 2021
Date of this review
03 Dec 2021 16:45:21
Reviewer 2 Report
This epidemiological study characterizes breast tumors on the basis
of their molecular profile (defined by the expression of hormone receptors and HER2) and the occurrence of metastases at diagnosis in relation to patient age in the Italian population using variables collected from population registries. The information presented in this paper may help to gain an understanding of breast cancer epidemiology in Italian population.
Author Response
Reviewer 2
(x) I would not like to sign my review report
( ) I would like to sign my review report English language and style
( ) Extensive editing of English language and style required
(x) Moderate English changes required
( ) English language and style are fine/minor spell check required
( ) I don't feel qualified to judge about the English language and style
|
Yes |
Can be improved |
Must be improved |
Not applicable |
|
|
Does the introduction provide sufficient background and include all relevant references? |
( ) |
(x) |
( ) |
( ) |
|
Is the research design appropriate? |
( ) |
(x) |
( ) |
( ) |
|
Are the methods adequately described? |
( ) |
(x) |
( ) |
( ) |
|
Are the results clearly presented? |
( ) |
(x) |
( ) |
( ) |
|
Are the conclusions supported by the results? |
( ) |
(x) |
( ) |
( ) |
Comments and Suggestions for Authors
This epidemiological study characterizes breast tumors on the basis of their molecular profile (defined by the expression of hormone receptors and HER2) and the occurrence of metastases at diagnosis in relation to patient age in the Italian population using variables collected from population registries. The information presented in this paper may help to gain an understanding of breast cancer epidemiology in Italian population.
We would like to thank the reviewer for reviewing our work and for their constructive suggestions.
Submission Date
22 October 2021
Date of this review
05 Dec 2021 08:03:06
Fine modulo
© 1996-2021 MDPI (Basel, Switzerland) unless otherwise stated